# *In vivo* identification and validation of novel potential predictors for human cardiovascular diseases

Omar T. Hammouda[1,2], Meng Yue Wu[1], Verena Kaul[1], Jakob Gierten[1,2¤], Thomas Thumberger[1], Joachim Wittbrodt[1]*

1 Centre for Organismal Studies Heidelberg, Heidelberg University, Heidelberg, Germany, 2 Heidelberg Biosciences International Graduate School, Heidelberg University, Heidelberg, Germany

¤ Current address: Department of Pediatric Cardiology and Congenital Heart Diseases, University Hospital Heidelberg, Heidelberg, Germany
* jochen.wittbrodt@cos.uni-heidelberg.de

**Data Availability Statement:** All relevant data are within the paper and its Supporting Information files.

## Abstract

Genetics crucially contributes to cardiovascular diseases (CVDs), the global leading cause of death. Since the majority of CVDs can be prevented by early intervention there is a high demand for the identification of predictive causative genes. While genome wide association studies (GWAS) correlate genes and CVDs after diagnosis and provide a valuable resource for such causative candidate genes, often preferentially those with previously known or suspected function are addressed further. To tackle the unaddressed blind spot of understudied genes, we particularly focused on the validation of human heart phenotype-associated GWAS candidates with little or no apparent connection to cardiac function. Building on the conservation of basic heart function and underlying genetics from fish to human we combined CRISPR/Cas9 genome editing of the orthologs of human GWAS candidates in isogenic medaka with automated high-throughput heart rate analysis. Our functional analyses of understudied human candidates uncovered a prominent fraction of heart rate associated genes from adult human patients impacting on the heart rate in embryonic medaka already in the injected generation. Following this pipeline, we identified 16 GWAS candidates with potential diagnostic and predictive power for human CVDs.

## Introduction

Genetics crucially contributes to the development and progression of cardiovascular diseases (CVDs), the global leading cause of death [1, 2]. Elevated resting heart rate in humans has been widely considered as a potential, modifiable risk factor of cardiovascular and all-cause mortality [3–6]. Since the majority of CVDs can be prevented by early intervention [7] there is a high demand for diagnostic and predictive CVD markers. Various model organisms have been previously employed in mutagenesis screens to identify and characterize relevant cardiac genes [8–12]. Alternatively, genome wide association studies (GWAS) on human patients correlate genes and CVDs after diagnosis and provide a valuable resource for those putative

**Funding:** fellowship of the Deutsches Zentrum für Herz-Kreislauf-Forschung (DZHK) to O.T.H Deutsche Herzstiftung e.V. (S/02/17) to JG. Add-On Fellowship for Interdisciplinary Science of the Joachim Herz Stiftung to JG. DFG CRC-1324 TP B4 to JW NIH 5R01ES029917 – 03, KiyosuTox to JW. The funders had no role in study design, data collection and analysis, decision to publish, or preparation of the manuscript.

**Competing interests:** The authors have declared that no competing interests exist.

causative genes with direct human/clinical relevance [13]. Indeed, previous efforts have been made to identify and validate candidate GWAS genes using *Drosophila* and zebrafish using RNAi and morpholinos, respectively [14]. However, there are still uncharacterized genes within the human GWAS candidates with no pre-existing evidence to the heart requiring experimental validation. The emergence of CRISPR technology has revamped genome editing [15] and consequently, functional gene validation in vertebrate models such as teleost fish [16–21]. However, there still is a lack of efficient gene targeting, and of high-throughput phenotyping pipelines allowing for the rapid and robust validation of candidate genes with implications for heart function. Recently, we demonstrated the power of targeted genome editing in the small animal model system medaka (*Oryzias latipes*) to validate trabeculation-associated genes [22]. The ease of manipulation combined with robust acquisition and analysis pipelines highlight the power of using fish embryos in high-throughput applications [23–26]. Embryos of fish model systems undergo extrauterine development in a transparent egg. This allows to monitor heart development and heart rate non-invasively in live undisturbed embryos for an extended period of time. Heart development, function and physiology in fish, though simpler, is in principle comparable to mammals [27–29]. Here we combined targeted genome editing via CRISPR/Cas9 [30, 31] with automated high-throughput imaging and heart rate analysis in isogenic medaka embryos [26] to enable functional analyses directly in the injected generation (F0). We tested the performance our assay with a positive control (*nkx2-5*), evaluated the random discovery rate and analyzed 40 heart phenotype-associated genes identified from human GWAS. Our assay uncovered that 57% of candidates assigned to human heart rate in GWAS also affected heart rate in fish embryos. We have thus experimentally validated understudied human GWAS candidates, identifying 16 genes with potential diagnostic and predictive power for human CVDs.

## Results

For the straight forward functional validation of GWAS candidates we aimed at combining CRISPR/Cas9-mediated targeted gene inactivation in the injected generation of medaka embryos with high content screening approaches to validate the impact of the loss-of-function on the heart rate (Fig 1A). We first assessed the efficiency of the CRISPR/Cas9 system by targeting a copy of *green fluorescent protein* (*gfp*) gene in a transgenic medaka reporter line expressing GFP and mCherry fluorescent proteins exclusively in the heart under the control of the *cardiac myosin light chain 7* (*myl7*) promoter. We employed the *heiCas9* mRNA variant, i.e. a Cas9 equipped with an early active nuclear localization signal enabling its immediate nuclear localization [31]. Injection of the *heiCas9* together with a guide RNA targeting *gfp* into medaka embryos at the 1-cell stage resulted in the complete loss of GFP expression in the heart (n = 8/8) (Fig 1B and S1B Fig). Only when injected later into a single blastomere of the four-cell stage or later, a mosaic pattern was observable. This demonstrates that the chosen heiCas9 acts uniformly in the injected cell and all of its descendants allowing functional analyses already in the injected generation.

For the validation of our assay, we investigated the loss-of-function of the cardiac-specific homeobox-containing transcription factor NKX2-5 as a positive control. In human patients, a single amino acid mutation in the homeodomain (R141C) was previously associated with atrial septal defect (ASD) and shown to cause delayed heart morphogenesis in adult mice [32]. To test our high-throughput imaging and heart rate analysis pipeline for functional *in vivo* gene validation in the injected generation, we targeted the region orthologous to R141C in medaka embryos using the CRISPR/Cas9 system [30, 31]. As a negative control, we targeted *oculocutaneous albinism* 2 (*oca2*) [33], a pigmentation gene unrelated to heart function. The bi-allelic

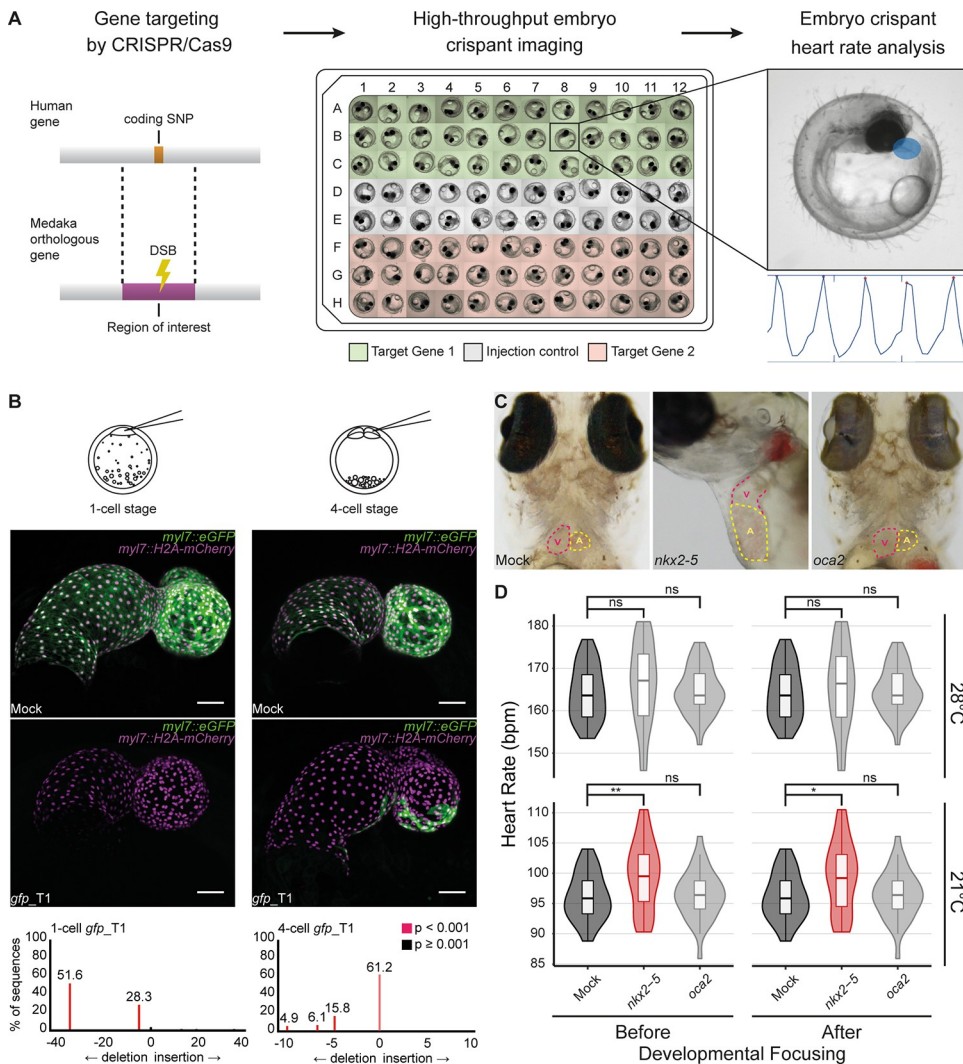

**Fig 1. Functional gene validation pipeline confirms heart rate phenotype in *nkx2-5* crispants.** (A) Schematic overview of our functional gene validation pipeline: position of human coding SNP mapped to medaka orthologous gene to define region of interest for CRISPR/Cas9 gene targeting (double strand break; DSB). 96-well plate layout of crispant embryos (Target Gene 1 and 2) separated by *GFP* mRNA mock-injected siblings. Embryos are subjected to high-throughput imaging followed by automated heart detection (blue area) and heart rate quantification (graphical output; *HeartBeat* software [26]. (B) Confocal images (mirrored) of GFP expression in mock-injected and *gfp* crispant embryo hearts of the *myl7::eGFP myl7::H2A-mCherry* reporter line (7 dpf). Embryos were injected either at the 1-cell or 4-cell stage. Note: complete loss of GFP expression when injected at the 1-cell stage (n = 8/8), while mosaic expression when injected at 4-cell stage (n = 4/4). Genotyping of *gfp* crispants display the genetic mosaicism resulting from Cas9-based targeting. Scale bars: 50 μm. For full image refer to S1 Fig. (C) Comparison of the atrium (A, dotted red line) and ventricle (V, dotted yellow line) in *GFP*-injected (Mock) and *nkx2-5* and *oca2* crispant embryos (9 dpf). Note: *nkx2-5* crispant shows dilated heart chambers while mock injected and *oca2* crispant embryos are indistinguishable. Loss of eye pigmentation in *oca2* crispants reflects high efficiency of knock-out rate in the injected generation. (D) Heart rate measurements (beats per minute, bpm) of *GFP*-injected (Mock; dark grey), *nkx2-5* and *oca2* crispant embryos (4 dpf) at 21 and 28°C, before (left) and after (right) exclusion of severely affected embryos (developmental focusing) reveal elevation of mean heart rates in *nkx2-5* targeted embryos, significant at 21°C (red). Significance was determined by two-tailed Student's t-test; *$p < 0.05$, **$p < 0.01$, ns (not significant; light grey). For biological replicates see Source Data Fig 1D in S1 Data.

editing of *oca2* and the subsequent loss of the eye pigmentation (with limited mosaicism) [34, 35] underscores the high efficacy of our system. Injections into wild-type medaka embryos were performed at the 1-cell stage, and hereafter resulting embryos are referred to as crispants

[20]. To address the impact on the heart rate we raised the medaka crispants until cardiac function was fully developed and the heart rate had reached a plateau at 4 days post fertilization [26] (4 dpf; developmental stage ~31–32 [36]).

To assess changes in mean heart rate with statistical significance, we took advantage of a 96-well plate format, and imaged multiple biological replicates of crispants of each targeted gene (3 rows; n = 36 per condition) as well as of *GFP mRNA* mock-injected siblings as internal plate control (2 rows; n = 24) (S2A Fig). The efficacy of heiCas9 under the given conditions was determined by targeting *oca2* as described above. We employed only sgRNAs that successfully target the desired loci (see Materials and Methods). To acutely assess heart function under different environmental conditions, embryos were acutely subjected to two different temperatures (21˚C and 28˚C) while imaging. The different temperatures act as environmental stressors to assess the heart rate response and to uncover phenotypes that would not be observed when imaging at a single set temperature. Heart rates of all embryos were quantitatively determined from the imaging data using the *HeartBeat* software [26], and randomly selected embryos were genotyped to correlate CRISPR/Cas9 targeting [35].

While mock or control (*oca2*) injected embryos did not show phenotypes, crispants of the positive control *nkx2-5* displayed a variety thereof. These ranged from global severe developmental delays to local cardiac malformations morphologically resembling the phenotypes previously observed in zebrafish *nkx2-5* mutants such as enlarged heart chambers (Fig 1C) [37]. Notably, in the negative control (*oca2* crispants) neither cardiac nor developmental phenotypes were observed (Fig 1C), indicating that targeting of *oca2*, injection and handling of the embryos as well as Cas9-mediated double-strand breaks did not impact on heart and general development *per se*. Quantitative heart rate comparison revealed an overall elevation in the mean heart rate of *nkx2-5* crispants (21˚C 99.7 bpm n = 34, 28˚C 166 bpm n = 34) compared to mock control siblings (21˚C 96.1 bpm n = 22, 28˚C 164 bpm n = 23) with a significant (p = 0.0074) difference at 21˚C (Fig 1D; left panel). Notably, independent experimental replicates targeting the same *nkx2-5* exon with two different sgRNAs robustly yielded a significant heart rate phenotype at 21˚C (S2B and S2C Fig). In contrast, the mean heart rate in *oca2* crispants was indifferent from mock control at either temperature (21˚C 96.4 bpm n = 35, 28˚C 165 bpm n = 35), validating *oca2* as bona fide negative control.

To avoid severe developmental delays in *nkx2-5* crispants to potentially skew heart rate comparisons, we further applied a developmental focusing filter. Only embryos having developed beyond stage 28 [36], at which cardiac function was previously shown to have reached a functional plateau [26], were chosen for statistical analysis. Developmental focusing, only excluded three embryos from the *nkx2-5* group which did not impact on the results (Fig 1D; right panel). These results underline the robustness of our pipeline and demonstrate its sensitivity to detect mild heart rate phenotypes reflecting cardiac function already at embryonic stages of medaka development.

Next, we determined the baseline probability of heart rate phenotypes by targeting a set of randomly selected genes with CRISPR/Cas9. From a total of 23622 annotated medaka coding genes in Ensembl [38], we used a random number generator to select 10 genes (Table 1). For each gene, a random exon was chosen for targeting via CRISPR/Cas9. Heart rates of 36 crispants were assessed per locus. To control for potential heart rate fluctuations in embryos within and across different experiments (i.e. 96-well plates), we included mock-injected siblings as internal plate control. Heart rates of target gene crispants and control siblings were scored and the means were compared before and after developmental focusing.

Comparative heart rate analysis revealed a heart rate phenotype in two out of the ten randomly selected genes at both temperatures measured (Fig 2, and S3 Fig). Remarkably, both genes of the random set, the *oxoglutarate dehydrogenase* (*ogdh*) and the *cell division control*

**Table 1. List of randomly selected genes.**

| Ensembl ID | Medaka Gene | Targeted exon (total exons) | Orthologous Human Gene |
|---|---|---|---|
| ENSORLG00000006335 | novel gene—*cdc42* | 5 (6) | na (*CDC42* by name) |
| ENSORLG00000000979 | novel gene—*ogdh* | 19 (22) | *OGDH* |
| ENSORLG00000005268 | *duox* | 17 (32) | *DUOX1* |
| ENSORLG00000007310 | *git2* | 1 (21) | *GIT2* |
| ENSORLG00000003492 | *mus81* | 8 (13) | *MUS81* |
| ENSORLG00000020766 | *or124-2* | 2 (2) | na |
| ENSORLG00000007400 | *plekha8* | 13 (13) | *PLEKHA8* |
| ENSORLG00000022757 | *ttl* | 2 (7) | *TTL* |
| ENSORLG00000005922 | *cabp4* | 2 (4) | *CABP2* |
| ENSORLG00000023106 | novel gene—*eml6* | 27 (36) | *EML6* |

Medaka Ensembl gene names, codes and exon targeted, as well as orthologous human genes as annotated in the 95th Ensembl release.

*protein 42 homolog (cdc42)* have been previously associated with heart phenotypes in human GWAS or were reported to play a role in heart function, respectively [39–41]. These results confirm the reliability of our assay to identify genes of a given set that affect cardiac function.

We next applied our pipeline to interrogate a larger, targeted selection of genes associated with cardiovascular diseases in human GWAS. We used GRASP [13], the genome-wide repository of associations between single nucleotide polymorphisms (SNPs) and phenotypes, to compile a list of 40 candidate genes from human GWAS with a coding association to heart phenotypes (hGWAS genes; Table 2). We focused on genes with no prior experimental link to

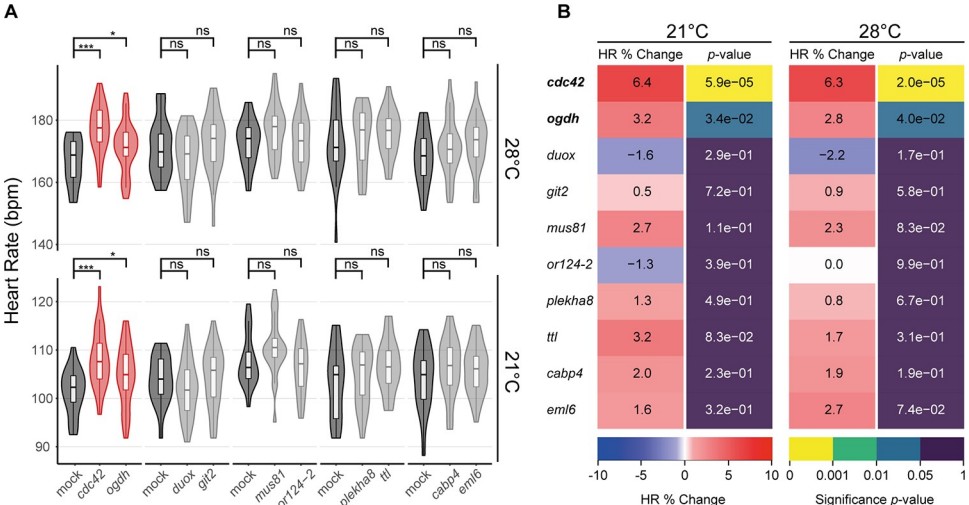

**Fig 2. Baseline probability of heart rate phenotype assessed via *in vivo* targeting of randomly selected genes.** (A) Heart rate measurements (beats per minute, bpm) of *GFP*-injected (Mock) and corresponding sibling crispant embryos (4 dpf) at 21 and 28°C after developmental focusing. Different experimental plates are represented by breaks on the x-axis. Significant differences in mean heart rates were determined between each crispant embryo group and its corresponding sibling control group by two-tailed Student's t-test; *p < 0.05, ***p < 0.001, ns (not significant). Crispants showing significant heart rate phenotype (red), *GFP*-injected controls (Mock; dark grey), crispants showing no significant heart rate phenotype (light grey). (B) Heatmap quantitative representation of the data shown in (A); for each measured temperature, the percent change in mean heart rate (HR % Change) between crispants and their corresponding control sibling, flanked by the statistical significance (*p*-value) of the observed change calculated by two-tailed Student's t-test on the full distribution in (A). Genes showing significant heart rate phenotypes are indicated in **bold**. For biological replicates see Source Data S3 Fig in S1 Data.

**Table 2. List of candidate genes extracted from human GWAS using GRASP 2.0 database.**

| Human Gene | Coding SNP ID | Associated heart phenotype | Association Reference | Medaka orthologue gene (Ensembl release) | Ensembl Gene Code |
|---|---|---|---|---|---|
| *ATP8B4* | **rs2452524** | **Pulse rate** | **[42]** | **atp8b4 (98)** | **ENSORLG00000005106** |
| *CASQ2* | **rs4074536** | **QRS interval** | **[43]** | **casq2 (89)** | **ENSORLG00000017885** |
| *CCDC141* | **rs17362588** | **Heart rate** | **[14]** | **na (89) (TBLASTN = ccdc141)** | **TBLASTN = ENSORLG00000030409** |
| *CEP85L* | **rs3734381** | **QRS interval** | **[43]** | **cep85l (91)** | **ENSORLG00000015455** |
| *CMYA5* | **rs10942901** | **Heart rate** | **[14]** | **cmya5 (91)** | **ENSORLG00000008983** |
| *COL9A1* | **rs592121** | **Pulse rate** | **[42]** | **col9a1b (98)** | **ENSORLG00000010431** |
| *GIGYF1* | **rs221794** | **Heart rate** | **[14]** | **gigyf1 (89)** | **ENSORLG00000003655** |
| *GRID2* | **rs1385405** | **Pulse rate** | **[42]** | **grid2 (98)** | **ENSORLG00000024663** |
| *HOMEZ* | **rs1055061** | **Sick sinus syndrome** | **[44]** | **homeza (89)** | **ENSORLG00000012220** |
| *KCNH2* | **rs1805123** | **QT interval** | **[45]** | **kcnh2 (98)** | **ENSORLG00000004137** |
| *MINAR1* | **rs2297773** | **Pulse rate** | **[42]** | **minar1 (98)** | **ENSORLG00000016707** |
| *MYRF* | **rs174535** | **RR interval** | **[46]** | **myrf (91)** | **ENSORLG00000006459** |
| *NACA* | **rs2926743** | **Heart rate** | **[14]** | **naca (98)** | **ENSORLG00000012246** |
| *OR5AU1* | **rs4982419** | **Pulse rate** | **[42]** | **na (98) (TBLASTN = no name)** | **TBLASTN = ENSORLG00000024679** |
| *PADI4* | **rs2240335** | **Pulse rate** | **[42]** | **na (98) (TBLASTN = padi2)** | **TBLASTN = ENSORLG00000007539** |
| *PPP1R9A* | **rs854524** | **Pulse rate** | **[42]** | **ppp1r9a (98)** | **ENSORLG00000004418** |
| *RNF207* | **rs846111** | **QT interval** | **[47]** | **rnf207b (91)** | **ENSORLG00000017207** |
| *SCN5A* | **rs1805126** | **QRS interval** | **[48]** | **na (89) (TBLASTN = scn4ab)** | **TBLASTN = ENSORLG00000003273** |
| *SSPO* | **rs10261977** | **Pulse rate** | **[42]** | **sspo (98)** | **ENSORLG00000004121** |
| *TRAPPC12* | **rs6767** | **Pulse rate** | **[42]** | **trappc12 (98)** | **ENSORLG00000017859** |
| *TTN* | **rs12476289** | **QT interval** | **[49]** | **ttn.2 (91)** | **ENSORLG00000018144** |
| *UFSP1* | **rs12666989** | **RR interval** | **[46]** | **na (89) (TBLASTN = ufsp1)** | **TBLASTN = ENSORLG00000022928** |
| *XYLB* | **rs17118** | **PR interval** | **[50]** | **xylb (91)** | **ENSORLG00000003755** |
| *ABCB1* | rs1128503 | Drug response CVD | [51] | abcb4 (91) | ENSORLG00000009269 |
| *BAG3* | rs3858340 | Sporadic dilated cardiomyopathy | [52] | bag3 (89) | ENSORLG00000013813 |
| *CLCNKA* | rs1805152 | Sporadic dilated cardiomyopathy | [52] | clcnk (89) | ENSORLG00000018693 |
| *CNOT1* | rs11866002 | Aortic valve calcium | [40] | cnot1 (91) | ENSORLG00000013734 |
| *EDN1* | rs150035515 | Aortic valve calcium | [53] | edn1 (89) | ENSORLG00000009276 |
| *HCN4* | rs529004 | Aortic valve calcium | [40] | hcn4 (91) | ENSORLG00000013180 |
| *MAML3* | rs11729794 | Congenital heart malformations | [54] | na (91) (TBLASTN = maml3) | TBLASTN = NCBI Ref Seq: XM_023954746.1 |
| *NUBP2* | rs344359 | LV systolic dysfunction | [55] | nubp2 (91) | ENSORLG00000007228 |
| *PIEZO1* | rs2290902 | Bicuspid aortic valve | [56] | piezo1 (91) | ENSORLG00000000402 |
| *PLG* | rs13231 | Aortic valve calcium | [40] | plg (91) | ENSORLG00000020532 |
| *RGS3* | rs12341266 | Hypertrophic cardiomyopathy | [56] | rgs3a (91) | ENSORLG00000006823 |
| *SCMH1* | rs10489520 | Ischemic stroke | [57] | scmh1 (91) | ENSORLG00000014207 |
| *SH2B3* | rs3184504 | Tetrology of fallot | [58] | sh2b3 (91) | ENSORLG00000003569 |
| *SLC17A3* | rs942379 | Bicuspid aortic valve | [56] | si:ch1073-513e17.1 (91) | ENSORLG00000007671 |
| *SMG6* | rs216193 | Aortic root size | [55] | smg6 (91) | ENSORLG00000003317 |
| *VEPH1* | rs1378796 | Sporadic dilated cardiomyopathy | [52] | veph1 (89) | ENSORLG00000012452 |
| *ZFHX3* | rs2228200 | Aortic valve calcium | [40] | zfhx3 (91) | ENSORLG00000007874 |

Human genes are categorized according to their association into "heart rate" (**bold**) and "non-heart rate" (non-bold) related phenotypes in human GWAS.

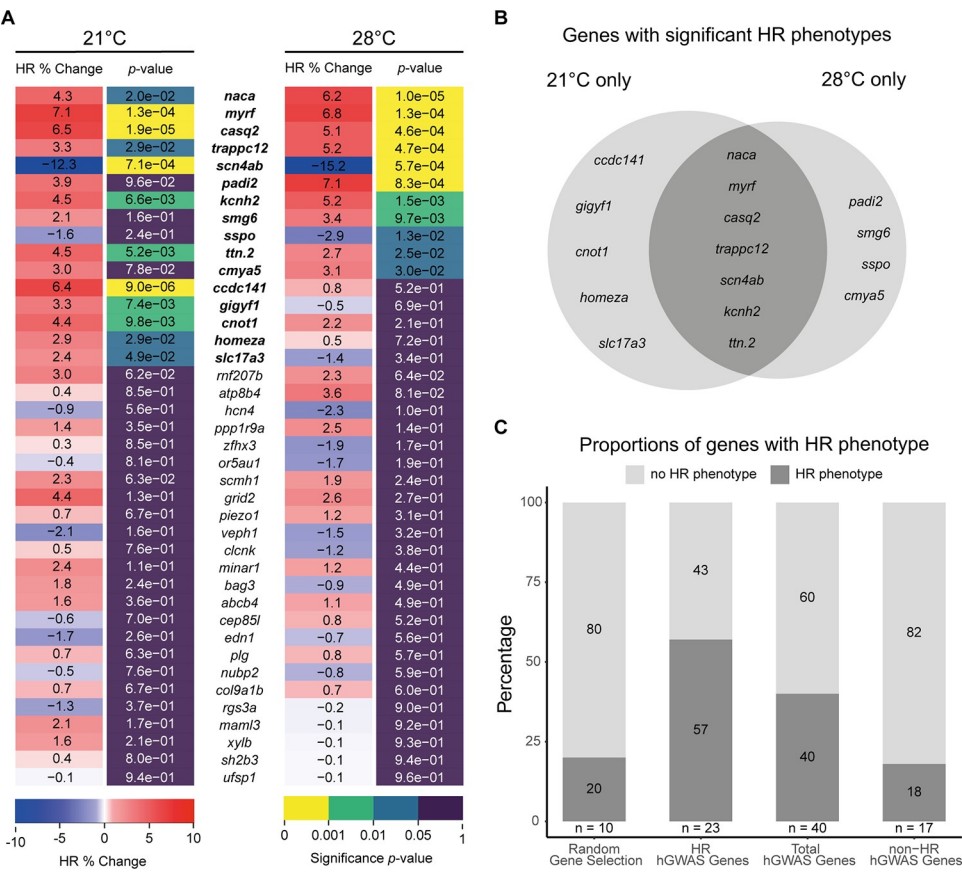

**Fig 3. Targeted human heart-GWAS validations reveal new genes affecting heart rate.** (A) Heatmap quantitative representation of the comparative heart rate analysis between each crispant embryo group and its corresponding control sibling group after developmental focusing (also see plots in S5 Fig); for each measured temperature, the percent change in mean heart rate (HR % Change) between crispants and their corresponding control sibling, flanked by the statistical significance (*p*-value) of the observed change calculated by two-tailed Student's t-test on the full distribution (S5 Fig). Genes showing significant heart rate phenotypes are indicated in **bold**. For biological replicates see Source Data S5 Fig in S1 Data. (B) Venn diagram summarizing the genes with significantly different heart rate (HR) phenotypes only at 21˚C, only at 28˚C or at both temperatures (dark grey). (C) Stacked plots representing percentage of genes showing a significant heart rate phenotype (dark grey) in each group. Number of genes for each group is denoted (n). hGWAS corresponds to the selection of genes associated to heart phenotypes in human GWAS.

heart function, while including few known heart genes as additional positive controls. To address the specificity of our approach, the selected hGWAS candidate genes were categorized according to their association into general heart phenotypes (n = 17) or specifically to heart rate phenotypes (n = 23; Table 2). Successful and efficient heiCas9 targeting of all candidate genes was assessed prior to experimentation by testing our sgRNAs *in vivo* in medaka embryos and confirmed by locus amplification followed by T7 Endonuclease I mismatch assay (Materials and Methods, S4 Fig).

Heart rates of candidate gene crispants and control injected siblings were scored and compared before and after developmental focusing (Fig 3A; S5 and S6 Figs). Across the hGWAS set of 40 genes, comparative heart rate analysis showed statistically significant heart rate phenotypes in a total of 16 genes (Fig 3B). The five positive controls, known to play key roles in heart functions such as cardiac contraction (*TTN* [59] and *NACA* [60]) and heart rate regulation (*CASQ2* [61], *KCNH2* [62] and *SCN5A* [63, 64]) clearly responded in the assay. Beyond known cardiac genes, we revealed new genes linked to various biological functions (*CCDC141*,

*GIGYF1*, *HOMEZ*, *MYRF*, *SMG6*, *CMYA5*, *CNOT1*, *SLC17A3*, *TRAPPC12*, *SSPO* and *PADI4*) which up to now, had little to no experimental evidence in cardiac function [65–68].

When looking at the candidates according to their GWAS association category ("heart rate" and "non-heart rate" phenotypes), we observed a pronounced positive correlation between the respective phenotypes observed in medaka crispants already at the embryonic stages and the associated phenotype in adult human GWAS. The proportion of heart rate-associated genes in hGWAS that yield a heart rate phenotype in medaka embryos (13/23) was elevated compared to the proportion of non-heart rate-associated genes yielding a heart rate phenotype (3/17). Even when considering the entire group, we observed a higher proportion of genes with an effect on heart rate in the targeted hGWAS gene set (16/40) compared to the randomly selected gene set (2/10) (Fig 3C). Taken together, phenotypes in early medaka embryos likely reflect risk factors in human adults, thus we uncovered functionally relevant heart rate phenotypes in previously uncharacterized genes.

In addition to the observed heart rate phenotypes in *trappc12* crispants, we also uncovered morphological heart phenotypes such as heart looping defects. Where in wild-type medaka, heart looping usually starts at around stage 27, when the atrium shifts to the right and lies adjacent to the ventricle [36], *trappc12* crispants revealed strong heart looping phenotypes (12/46) not observed in *oca2* crispants (0/22) or mock-injected embryos (0/26) (Fig 4A).

In crispants of *scn4ab*, heart rate analysis interestingly uncovered a bimodal distribution, with a population displaying roughly half the average heart rate at both recorded temperatures (Fig 4B). Visual inspection of the *scn4ab* crispant embryos revealed an arrhythmic heart anomaly similar to previous reports in zebrafish *scn5a* mutants [63], i.e. ventricular beat skipping, reminiscent of a clinically relevant atrio-ventricular block (AV-block) arrhythmia in humans. Scoring the beat frequency of both heart chambers separately in individual embryos exposed the impaired rhythm of atrial to ventricular contractions, which resulted in a delay or even skipping of ventricular beats in the *scn4ab* crispants but not in control siblings (Fig 4C). *scn4ab* crispants displayed various severities of the arrhythmia from mild (regular heart beats with occasional beat skipping), to moderate (consistent 2:1 atrial to ventricular contraction; S1 Movie), to severe (3:1 or more; S2 Movie). Interestingly, even heavily affected *scn4ab* crispants survived until hatching. Impressively, the prevalence of the arrhythmia phenotype in *scn4ab* crispants was markedly high, exceeding 90% of the injected embryos, once more reflecting the high efficiency of the *heiCas9* and the high penetrance of the mutations introduced. Notably, the arrhythmia phenotype of *scn4ab* mutants bred to homozygosity did not differ from the phenotype observed in the injected generation F0 (S1 Movie), verifying the specificity of the phenotype. These results further underscore the efficacy and reliability of medaka F0 crispant analysis as a rapid validation tool to identify genes with a functional link to human cardiac diseases.

## Discussion

Most cardiovascular diseases can be prevented if diagnosed and treated early. Previous studies have shown the importance of the resting heart rate as a vital risk factor both in terms of prediction and prevention of CVDs [3, 6, 69]. An increase of 5 beats per minute correlates with a 20% increase in risk of mortality [69], and reducing the resting heart rate has proven to improve the clinical outcomes of various CVDs [4, 6]. Thus, the heart rate poses as an important 'modifiable' risk factor that could allow prediction or early diagnosis and therefore early intervention, potentially preventing onset of CVDs.

Human GWAS have been performed in search of genetic determinants of CVDs, and although a wide array of candidate genes with various functions are being associated to heart

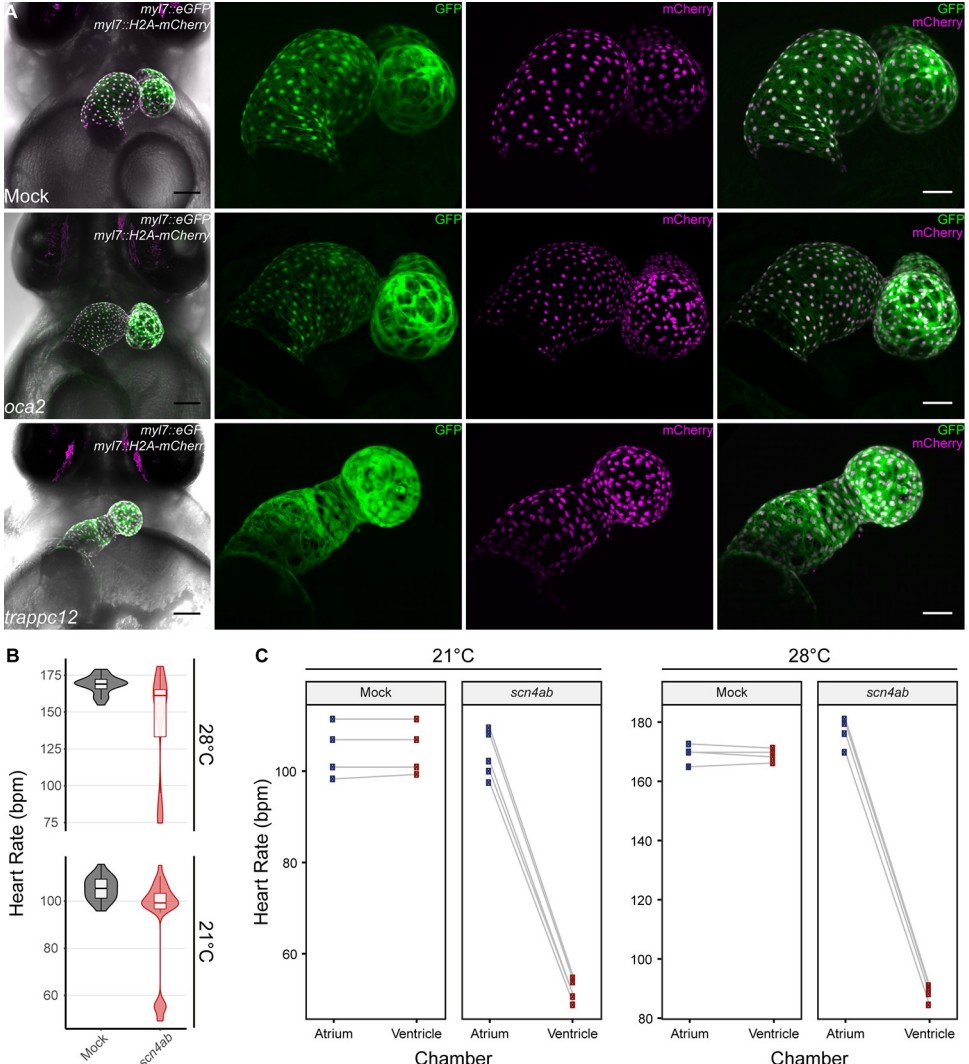

**Fig 4. Heart looping defects and cardiac arrhythmia in medaka crispant embryos.** (A) Confocal images (mirrored) of hearts of mock-injected, *oca2* and *trappc12* crispants of the *myl7::eGFP myl7::H2A-mCherry* reporter line (7 dpf); note the heart looping defect observed in *trappc12* crispants. Scale bars: 100 μm (First panel on left) and 50 μm (blow-up images). (B) Heart rate measurements (beats per minute, bpm) of *GFP*-injected (Mock; dark grey; n = 22) and *scn4ab* crispant (red; n = 32) embryos (4 dpf) at 21 and 28˚C; note the bimodal distribution in *scn4ab* crispants. (C) Paired plots showing heart rate scores for each chamber separately (atrium in blue; ventricle in red) of individual embryos from B at both temperatures.

phenotypes in human GWAS, further focus is usually turned to those few genes with pre-existing indication of cardiac function. There is still an unaddressed blind spot of linked genes with no prior connection to cardiac development or function. Thus, it is important to address the role of such genes in heart function through experimental validation in model organisms, in pursuit of novel causative genes for CVD diagnosis. Gene validation attempts have been undertaken using invertebrate models (e.g. drosophila using RNAi) [14, 70, 71] which quickly provide insights but still require validation and translation in a vertebrate model. Teleost models offer a compromise between throughput and translational relevance. Before genome targeting approaches were available, morpholino-based knock-down approaches have been used in zebrafish for high throughput genetic screening [11, 14]. Currently available CRISPR/Cas9

technology, on the other hand, allows immediate, highly efficient gene knock-outs at much lower costs [16–21]. Here we apply a high-throughput heart rate imaging and analysis pipeline coupled to a reverse genetic validation approach via highly efficient CRISPR/Cas9-mediated mutagenesis in a genetically suited vertebrate model (Fig 1A) to validate genetically linked but understudied candidate genes.

F0 mutagenesis screens are becoming more and more relevant [21, 25, 72, 73], largely due to the improvements in CRISPR/Cas9 gene targeting efficiency by modification of the enzyme or promoting nuclear localization [31]. Overall, gene targeting in medaka using CRISPR/Cas9 has proven to be highly efficient, as shown by the prominent loss of GFP expression in *gfp* crispants when injected at the 1-cell stage (Fig 1B and S1 Fig) and the prominent bi-allelic mutagenesis as apparent by the loss of eye-pigmentation in the *oca2* crispants (Fig 1C and S2A Fig). A similarly high penetrance was also observed in the *scn4ab* crispants, where we detected and quantified severe arrhythmia phenotypes with our assay (Fig 4B and 4C, S1 and S2 Movies). A 90% prevalence of the arrhythmia phenotype and an absence of global phenotypes further reflect the specificity of this phenotype.

A subset of the target genes in our assay (e.g. *nkx2-5*, *smg6*, *naca*, *ttn.2*, *abcb4*) however, yielded a rather broad range of global developmental phenotypes, reflecting their important roles already early during embryonic development. To address this un-avoidable outcome when tackling genes with broader function (e.g. transcription factors or essential genes), we applied a developmental focusing filter in the analysis phase. Doing so, we avoid a biased assessment of the heart rate by ensuring the comparability of the crispant embryos on a global developmental scale, which in turn allows emphasizing cardiac-specific effects. Interestingly however, developmental focusing, although deemed important, in only few cases significantly altered the outcome of the analysis (S6 Fig). This reflects the robustness of the assay and the homogeneity of CRISPR/Cas9-induced phenotypes in the isogenic background of the medaka line used.

In a set of ten randomly chosen genes we observed a baseline occurrence of heart-affecting genes of about 20% (Fig 2 and S3 Fig). Relevantly, both genes had been implicated in heart function already, further highlighting the reliability of our model and approach. For *cdc42*, there is *a priori* evidence of its human orthologue in heart development as well as in regulating heart function across species [39, 41]. Surprisingly, we did not find any associations (coding or non-coding) of *CDC42* to heart phenotypes in human GWAS according to the GRASP database [13]. As for *ogdh*, no experimental evidence in cardiac function had been previously reported, but a polymorphism located on one of its exons has been associated to heart phenotypes in human GWAS [40]. Except for *duox*, which has been reported as having an indirect role in cardiac regeneration in zebrafish [74], none of the other randomly selected genes are so far connected with cardiac function. In summary, from the random gene set, the only two affecting the heart beat (*ogdh* and *cdc42*) are connected to cardiac function, one of which (*CDC42*) by prior evidence [39, 41].

All but one positive (*HCN4*) controls among the hGWAS candidates resulted in a pronounced heart rate phenotype in our assay, reflecting their role in cardiac contraction (*TTN* [59] and *NACA* [60]), cardiac conduction and heart rate regulation (*CASQ2* [61], *KCNH2* [62] and *SCN5A* [63, 64]). In case of the missing heart rate phenotypes anticipated in *hcn4* crispants, we suspect compensation by its paralog *hcn4l*.

For eleven hGWAS candidate genes, our analysis provided the first experimental evidence validating a cardiac function, and accordingly put these identified genes under the spotlight as new targets for future in-depth characterization and as candidates for the prediction of heart diseases prior to their onset. This is impressively substantiated by the emerging studies on the CCR4-NOT (*CNOT1*) complex in heart structure and function [66, 67].

Throughout the study, we primarily focused on the heart rate as a measure of cardiac function due to its ease of quantification and interpretation in high-throughput. However, we did also notice heart morphological phenotypes in some crispants across our hGWAS candidates list. Among others, *trappc12* crispants showed heart looping defects, resulting in the improper placement of the atrium compared to the ventricle (Fig 4A).

Grouping the candidate genes according to their heart phenotype GWAS association into heart rate and non-heart rate-related phenotypes further exposed the prominent positive correlation between the associated human phenotype and the observed phenotype in medaka (Fig 3C). Medaka's isogenic background, a product of inbreeding over multiple generations [75], allowed the detection of subtle changes in heart rate immediately in F0 crispants. This accelerated the analysis and avoided the necessity to analyze homozygous offspring in the second and third generation after CRISPR targeting.

It is noteworthy that despite the evolutionary distance from fish to humans, the medaka phenotypes match the class of hGWAS effects. This is even more relevant since the roles of the genes in medaka were validated at early larval stages, suggesting that the validated marker genes have predictive power in humans. This deep functional conservation emphasizes the potential of our approach for the identification and validation of novel predictive genetic markers for cardiovascular diseases in humans. We have showcased a highly versatile, sensitive and robust high-throughput reverse genetic validation assay to address the pool of understudied putative candidates.

Considering the bottleneck in the analysis pipeline, future advances in artificial intelligence (AI)-based image analysis coupled to high-throughput imaging platforms will allow the automated multidimensional feature extraction in high throughput. With this upscale of gene validation, a more complete understanding of the genetic factors involved in heart function seems within reach. In the future, the combination of genetic validation and drug screening in a single platform building on our assay will facilitate the simultaneous identification of novel genetic players and interacting small molecules with rescuing power.

## Materials and methods

### Ethics statement

All fish are maintained in closed stocks at Heidelberg University. Medaka (*Oryzias latipes*) husbandry (permit number 35–9185.64/BH Wittbrodt, Regierungspräsidium Karlsruhe) was performed according to local animal welfare standards (Tierschutzgesetz §11, Abs. 1, Nr. 1) in accordance with European Union animal welfare guidelines [76]. The fish facility is under the supervision of the local representative of the animal welfare agency. The following medaka stocks and transgenic lines were used: wild-type Cabs and *myl7*::*eGFP myl7*::*H2A-mCherry* transgenic HdrR-II strain. Medaka embryos were used at stages prior to stage 42. Medaka were raised and maintained as described previously [77].

### Generation of the transgenic *myl7* dual-reporter line

For dual-color cardiac imaging, *myl7*::*eGFP* and *myl7*::*H2A-mCherry* transgenic medaka lines were generated in the wild-type HdrR-II background. A modified version of pDestTol2CG (http://tol2kit.genetics.utah.edu/index.php/PDestTol2CG) was used containing a *myl7*::*eGFP* reporter cassette. For the second, nuclear reporter, the *eGFP* was replaced by an *H2A-mCherry* insert. Both plasmids were each co-injected at 10 ng/μl with 10 ng/μl Tol2 transposase mRNA into HdrR-II one-cell stage embryos using the microinjection technique as previously described [78] to generate separate reporter lines. A double transgenic line was derived from a

cross of the *myl7::eGFP* line to *myl7::H2A-mCherry* line and maintained for CRISPR-Cas9 injections.

## Candidate gene selection

For the unbiased gene targeting, an online random number generator was used to generate 10 numbers between 1 and 23622, corresponding to the number of annotated medaka coding genes in Ensembl [38] (Table 1). The number of exons for each gene was counted and a random number was generated to select the exon for CRISPR/Cas9 targeting. For the targeted human heart-GWAS (hGWAS) gene selection, the genome-wide repository of associations between SNPs and phenotypes (GRASP v2.0) was used [13]. In the search field, "Heart" and "Heart rate" were chosen as the respective categories for all heart- and heart rate-related phenotypes associated in human GWAS, only coding SNPs (i.e. SNP functional class = exons) were searched for. List of resulting genes was extracted (Table 2), and candidate genes for the functional validation assay were chosen. The focus was on uncharacterized genes, or genes with no prior experimental link to heart function, yet some known heart genes were included as proof of concept. For each hGWAS candidate gene, the corresponding medaka ortholog was extracted using Ensembl [38]. For the few genes which did not have an annotated medaka ortholog, the human protein sequence was BLASTed using the "tblastn" function of the NCBI BLAST (https://blast.ncbi.nlm.nih.gov/Blast.cgi) and Ensemble (http://www.ensembl.org/Multi/Tools/Blast) online tools to obtain a target medaka locus. Using Geneious 8.1.9 (https://www.geneious.com), regions of interest (ROI) on medaka orthologous genes for CRISPR targeting were primarily chosen based on the corresponding location of human SNP when aligning the medaka and human protein sequences.

## sgRNA target sites selection and *in vitro* transcription

All sgRNA target sites used in this study are listed in S1 Table. sgRNAs were designed with CCTop as described in Stemmer et al. [30]. sgRNA target sites were selected based on number of potential off-target sites and their corresponding mismatches. Preferably, sgRNAs selected had no off-target site or at least 3 nucleotide mismatches. sgRNA for *oca2* was the same as in Lischik et al. [34]. Cloning of sgRNA templates and *in vitro* transcription was performed as detailed in Stemmer, et al. [30]. All sgRNAs were initially tested after synthesis for *in vivo* targeting via injections into medaka embryos, followed by genotyping using our filter-in-tips protocol [35], in brief terms, by PCR amplification of target locus followed by T7 Endonuclease I assay (New England Biolabs) (S4 Fig).

## Microinjection

Medaka one-cell or four-cell stage embryos were injected into the cytoplasm as previously described [30]. Injection solutions for CRISPR targeting comprised: 150 ng/μl *heiCas9* mRNA [31], 15 ng/μl respective sgRNA and 10 ng/μl *GFP* mRNA as injection tracer. Control siblings were injected with 10 ng/μl *GFP* mRNA only. Injected embryos were incubated at 28˚C in embryo rearing medium (ERM), screened for GFP expression at 1 dpf and transferred to methylene blue-containing ERM (or plain ERM for reporter lines) and incubated at 28˚C until heart rate analysis (4 dpf) or confocal microscopy (7 dpf).

## Sample preparation and imaging

For the heart rate assay, one day prior to imaging (3 dpf), medaka embryos were transferred from methylene blue-containing ERM into plain ERM and incubated at 28˚C. On day of

imaging (4 dpf), individual medaka embryos (36 per sgRNA and 24 control injected) were administered to a 96 U-well microtiter plate (Nunc, Thermofisher #268152) containing 200 µl ERM per well and sealed using gas-permeable adhesive foil (4titude, Wotton, UK, 4ti-0516/96). Plates were automatically imaged using an ACQUIFER Imaging Machine (DITABIS AG, Pforzheim, Germany) at 21 and 28˚C with a 30-minute equilibration period before each measurement. Images were acquired in brightfield using 130 z-slices (dz = 0 µm) and a 2x Plan UW N.A. 0.06 objective (Nikon, Düsseldorf, Germany) to capture the centered embryo. Integration times were fixed with 80% relative white LED intensity and 10 ms exposure time. Therefore, the whole 96-well plate was captured, with image sequences (videos) of entire microwells of approx. 10 seconds with 13 frames per second (fps). More details can be found in Gierten, et al. [26].

For the live confocal microscopy of the reporter lines, injected embryos were treated from 4 dpf onwards with 5x phenylthiourea (PTU) in ERM solution to prevent pigmentation. On the day of imaging (7 dpf), PTU solution was washed away with ERM, embryos were rolled on fine sand paper and de-chorionated by incubation in hatching enzyme. Following de-chorionation, embryos were treated with 50 mM 2,3-butanedione monoxime (BDM) in 1x Tricaine solution until de-coupling of heart beat (~40 mins), which resulted in fully dilated heart chambers. Embryos were mounted ventrally on Matek dishes in 1.5% low-melting agarose with 85 mM BDM in 3x Tricaine solution. To avoid dehydration, mounted samples were covered with 30 mM BDM in 1x Tricaine solution throughout the imaging session. All confocal microscopy images were acquired at a Leica TCS SP8 with 10x dry or 20× oil objective, z-stacks of 200–300 µm were acquired with a z-step of 5 µm or 1 µm for 10x and 20x acquired images, respectively.

## Heartbeat detection and data analysis

Image optimizations prior to analysis, as well as heart rate analysis using the *HeartBeat* software were performed as previously described [26]. In some instances, heart rates could not be scored due to inconvenient embryo orientations shielding the view of the heart. For *scn4ab* crispants with cardiac arrhythmias, atrium and ventricle for individual embryos were separately segmented, and the respective beating frequency for each chamber was measured. Data plots were generated using ggplot2 package [79] in R 3.6.1 [80] and R-studio 1.2.1335 [81]. Statistical analysis for heart rate comparisons were computed in R. Significant differences were determined by two-tailed Student's t-test. Significant *p*-values are indicated with asterisks (*) with $*p < 0.05$, $**p < 0.01$, $***p < 0.001$ and ns (not significant). Maximum intensity projections of confocal microscopy images were processed via Fiji image processing software.

## Embryo genotyping

Nucleic acid extraction and genotyping of embryos was done as previously described [35]. Briefly, after imaging, embryos in 96-well plate were lysed in 50 µl Milli-Q water + 50 µl FinClip lysis buffer each (0.4 M Tris-HCl pH 8.0, 5 mM EDTA pH 8.0, 0.15 M NaCl, 0.1% SDS in Milli-Q water) using a custom 96-well mortar. The mortar was pre-cleaned by incubation in hypochlorite solution (1:10 dilution of commercial bleach reagent) for at least 15 minutes followed by 5 minutes incubation in Milli-Q water. Plates containing lysed embryos were stored at 4˚C until genotyping. To confirm CRISPR on-target activity, per experimental plate, 2 embryos per condition were chosen at random for genotyping by PCR amplification of target locus using our filter-in-tips approach [35], followed by T7 Endonuclease I Assay (New England Biolabs) (S4 Fig). 30 PCR cycles were run in all samples, all primers used for PCR are listed in S2 Table. Annealing temperatures were calculated using the online NEB Tm calculator (https://tmcalculator.neb.com/).

## Supporting information

**S1 Fig. Highly efficient CRISPR/Cas9-mediated editing in injected (F0) generation.** (A) Distribution of reporter expression in mock injected and *gfp_T1* crispants of the *myl7*::*eGFP myl7*::*H2A-mCherry* reporter line (4 dpf). Embryos were injected either at the 1-cell or 4-cell stage. Note: complete lack of GFP-expressing embryos when injected at the 1-cell stage. Biological replicates for each group is denoted (n). (B) Confocal images (mirrored) of GFP expression in mock-injected and *gfp* crispant embryo hearts of the *myl7*::*eGFP myl7*::*H2A-mCherry* reporter line (7 dpf). Embryos were injected either at the 1-cell or 4-cell stage. Note: complete loss of GFP expression when injected at the 1-cell stage (n = 8/8), while mosaic expression when injected at 4-cell stage (n = 4/4). Scale bars: 100 μm (First panel on left) and 50 μm (blow-up images).
(TIF)

**S2 Fig. Consistent heart rate phenotype observed in medaka nkx2-5 crispants.** (A) Overview of 96-well plate with embryos (4 dpf) injected with sgRNA against *nkx2-5* or *oca2*, as well as embryos mock injected with *GFP mRNA* (Fig 1D). Note the loss of eye pigmentation in *oca2* crispant embryos. (B-C) Heart rate measurements of *GFP*-injected (Mock; dark grey) and *nkx2-5* crispant embryos (4 dpf) ((B) second replicate of *nxk2-5_T4*; (C) different sgRNA *nkx2-5_T5* targeting same region of interest) at 21 and 28˚C, before and after exclusion of severely affected embryos (< stage 28; developmental focusing). Significant differences are shown in red and were determined by two-tailed Student's t-test; $*p < 0.05$, $**p < 0.01$, ns (not significant; light grey). For biological replicates see Source Data S2 Fig in S1 Data.
(TIF)

**S3 Fig. Developmental focusing does not alter analysis outcome of random gene selection.** Heatmap quantitative representation of the comparative heart rate analysis between each crispant embryo group and its corresponding control sibling group before and after developmental focusing; for each measured temperature, the percent change in mean heart rate (HR % Change) between crispants and their corresponding control sibling, flanked by the statistical significance (*p*-value) of the observed change calculated by two-tailed Student's t-test on the full distribution. Genes showing significant heart rate phenotypes are indicated in **bold**. For biological replicates see Source Data S3 Fig in S1 Data.
(TIF)

**S4 Fig. Confirmation of CRISPR-mediated in vivo gene editing via T7EI mismatch assay.** Representative examples of validated CRISPR-mediated gene targeting *in vivo*, confirming successful heiCas9 targeting and cleavage via sgRNAs employed. PCR amplification of target locus followed by T7EI mismatch cleavage assay (T7EI) demonstrates successful *in vivo* gene editing of target genes yielding a heart rate phenotype (red) as well as genes not yielding a heart rate phenotype (black). Per sgRNA, two randomly selected individual embryos were genotyped, while including a negative control (water; -) as well as a mock-injection control with GFP mRNA only (mock).
(TIF)

**S5 Fig. Comparative analysis of mean heart rates in targeted hGWAS gene selection.** Heart rate measurements (beats per minute, bpm) of *GFP*-injected (Mock; dark grey) and corresponding sibling crispant embryos (4 dpf) at 21 and 28˚C after developmental focusing (also see heatmap representation of the data in Fig 3A). Different experimental plates are represented by breaks on the x-axis. Significant differences in mean heart rates were determined between each crispant embryo group and its corresponding sibling control group by two-tailed

Student's t-test; $^*p < 0.05$, $^{**}p < 0.01$, $^{***}p < 0.001$, ns (not significant). Red groups correspond to crispants showing significant heart rate phenotypes, and light grey groups correspond to crispants showing no significant heart rate phenotype. For biological replicates see Source Data S5 Fig in S1 Data.
(TIF)

**S6 Fig. Developmental focusing does not alter analysis outcome of targeted hGWAS genes.** Heatmap quantitative representation of the comparative heart rate analysis between each crispant embryo group and its corresponding control sibling group before and after developmental focusing; for each measured temperature, the percent change in mean heart rate (HR % Change) between crispants and their corresponding control siblings, flanked by the statistical significance (*p*-value) of the observed change, calculated by two-tailed Student's t-test on the full distribution. Genes showing significantly different heart rate phenotypes are indicated in **bold**. For biological replicates see Source Data S5 Fig in S1 Data.
(TIF)

**S1 Movie. Moderate AV-block arrhythmia observed in medaka F0 scn4ab crispants and homozygous F2 mutants.** Side by side comparison of rhythmic heartbeat of *GFP*-injected (Mock; left) and arrhythmic *scn4ab* crispants (F0; middle) as well as homozygous mutants (F2; right) displaying 2:1 AV-block phenotype. Videos of medaka embryos (5 dpf) were acquired using a stereomicroscope under bright field illumination.
(AVI)

**S2 Movie. Severe AV-block arrhythmia observed in medaka scn4ab crispants.** Side by side comparison of rhythmic heartbeat of *GFP*-injected (Mock; left) and arrhythmic *scn4ab* crispants displaying severe AV-block phenotype (right). Videos of medaka embryos (9 dpf) were acquired using a stereomicroscope under bright field illumination.
(MP4)

**S1 Table. List of sgRNAs used.**
(DOCX)

**S2 Table. List of primers used for genotyping by PCR.**
(DOCX)

**S1 Data.**
(DOCX)

## Acknowledgments

We thank A. Cornean for providing the *gfp*_T1 guide RNA and for his help in acquiring the confocal images. We thank V. Weinhardt, E. Tsingos and all members of the Wittbrodt lab for their critical, constructive feedback on the procedure and the manuscript, F. Loosli and S. Lemke for their constructive feedback towards the project as well as J. Backs and E. Furlong for an outside perspective. We thank T. Kellner for excellent technical support. We acknowledge the excellent fish husbandry of E. Leist, M. Majewsky and A. Saraceno. We thank J. Gehrig (ACQUIFER Imaging GmbH) for supporting us with the Imaging machine.

## Author Contributions

**Conceptualization:** Omar T. Hammouda, Thomas Thumberger, Joachim Wittbrodt.

**Data curation:** Omar T. Hammouda.

**Formal analysis:** Omar T. Hammouda, Meng Yue Wu, Verena Kaul.

**Funding acquisition:** Joachim Wittbrodt.

**Investigation:** Omar T. Hammouda.

**Methodology:** Omar T. Hammouda, Thomas Thumberger, Joachim Wittbrodt.

**Project administration:** Joachim Wittbrodt.

**Resources:** Jakob Gierten, Joachim Wittbrodt.

**Supervision:** Joachim Wittbrodt.

**Validation:** Omar T. Hammouda.

**Visualization:** Omar T. Hammouda.

**Writing – original draft:** Omar T. Hammouda, Joachim Wittbrodt.

**Writing – review & editing:** Omar T. Hammouda, Thomas Thumberger, Joachim Wittbrodt.

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
