## [Decision Letter · Decision Letter 0]

24 Sep 2021

PONE-D-21-24196*In vivo* identification and validation of novel potential predictors for human cardiovascular diseasesPLOS ONE

Dear Dr. Wittbrodt,

Thank you for submitting your manuscript to PLOS ONE.  We feel the manuscript is technically sound and has merit but both reviewers have raised important points needed to be addressed. Therefore, we invite you to submit a revised version of the manuscript that addresses the points raised during the review process.

We look forward to receiving your revised manuscript.

Kind regards,

Gaurav Varshney, Ph.D.

Academic Editor

PLOS ONE

Journal Requirements:

Reviewers' comments:

Reviewer's Responses to Questions

**Comments to the Author**

1. Is the manuscript technically sound, and do the data support the conclusions?

Reviewer #1: Partly

Reviewer #2: Yes

2. Has the statistical analysis been performed appropriately and rigorously? 

Reviewer #1: Yes

Reviewer #2: I Don't Know

3. Have the authors made all data underlying the findings in their manuscript fully available?

Reviewer #1: Yes

Reviewer #2: Yes

4. Is the manuscript presented in an intelligible fashion and written in standard English?

Reviewer #1: Yes

Reviewer #2: Yes

5. Review Comments to the Author

Reviewer #1: The article by Hammouda et al. describes functional validation of candidates genes identified for human cardiovascular diseases through GWAS studies in Medaka embryos. Authors layout a plan where they first identified the candidate genes from human GWAS data. Then they identified the appropriate Medaka homolog for the gene and, using CRISPR/Cas9, targeted the gene. Surviving embryos were allowed to grow for four days when they had a stable heartbeat/rate and then imaged in a 96 well format dish with changes in the heart rate compared to the controls. Conceptually it is a nice design of experiments to validate candidate gene function in fish embryos. Still, I am concerned that it should not be a misused approach/technique similar to morpholinos. I have the following concerns:

1: All of the analysis is carried out in F0 animals which are mosaics, and it is hard to predict the cutting/targeting efficiency for each gRNA. Authors should provide additional evidence on cutting efficiencies for their most potent and weakest phenotype carrying animals and rule out the phenotypes/changes in heart rate are not mosaicism based.

2: Temperature-dependent changes in the heart rate of CRISPR animals are a bit of an issue. Many GWAS-linked genes need a "modifier" to set the phenotype; mimicking those conditions in fish embryos by modifying temperature is not the best way. Fish embryo development is highly plastic, and it slows down or accelerates based on how cold or warm the growing conditions are. Perhaps authors should consider combinatorial experiments where Crispants for 2 or 3 genes show more severe phenotypes than the single gene alone.

3: In Fig 3C (Bar 3), 40% of randomly selected GWAS genes showed HR phenotype. 57% of HR-specific GWAS genes showed heart phenotype. The sample size for this population is almost half compared to total hGWAS genes. This raises the concern about how effective the method is in deciphering correct information.

Overall, additional work is needed before this can be used as a tool to validate human gene function in fish embryos.

Reviewer #2: In the submitted manuscript, Hammouda and colleagues outline the use of medaka F0 assays to uncover congenital heart disease genes from GWAS studies and other patient-derived data. The expansion of accessible CRISPR-Cas9-based mutagenesis has rendered such approaches increasingly feasible, yet concerted efforts to establish platforms, protocols, workflows, and coordinated testing of candidate genes are in great need for a variety of diseases. The outlined approach by Hammouda and colleagues is therefore timely and an interesting addition to already existing work.

The manuscript is overall technically sound and well-executed, as common for the authors' lab. The work does address an important aspect of disease gene discovery and definitely warrants publication. However, the manuscript is presented in a "medaka bubble", i.e. not considering work that has been ongoing in the past two decades in mouse as well as in zebrafish. While clearly and undisputedly of interest, the paper phrases the issue of CHD association screening as previously neglected or not addressed, which is misleading. The reviewer doesn't think the authors need to render their own (and exciting) work more novel as it is by starting a narrative that their work is filling a so-far underappreciated gap that is uniquely possible in medaka.

With a little bit of editing, the reviewer has no doubt that the manuscript will be well-received in the model organism community after publication.

* Major Points:

1) From the onset, the authors frame their manuscript without any context to the tremendous amount of technical and experimental work that has already been done in the field of discovering CHD genes. The authors refer to their work as addressing a blind spot (i.e. line 246), which is defining an incorrect narrative.

At the very least, the authors should acknowledge previous work that puts their own work in context, i.e. as complementary approach. Large screens have been performed in mice to correlate found mutants with human CHD data (i.e. 10.1093/hmg/ddq211; 10.1161/CIRCIMAGING.113.000451) and are still ongoing. Further, in zebrafish, extensive experiments have been previously performed to characterize cardiac defects (i.e. 10.1242/dev.099796). Further, the entire approach of using F0 crispants for phenotype readouts, as originally developed in zebrafish, is sidelined and the involved caveats not mentioned (see also below (10.1371/journal.pone.0098186; 10.1101/gr.186379.114; 10.1242/dev.134809 that defines the term crispants; or newest developments such as 10.7554/eLife.59683).

2) The authors chose 10 random genes for targeting as validation (line 135ff). While an interesting approach, the authors then also chose a random exon. That seems like a possibly problematic approach - targeting last exons close to the stop codon could lead to merely hypomorphic or still functional alleles, alternative promoter use could lead to transcripts that don't include mutated exons, etc. The authors should provide further info on guide position (i.e. schematics?) in addition to Table 1 and the used guide sequences.

3) Overall, the authors focus on cardiac rhythm as principal readout, which is an exciting and highly disease-relevant phenotype. However, again, the authors phrase their overall manuscript as aimed at discovering heart disease genes. The authors are encouraged to rephrase their storyline as heart rhythm-focused, which is of high merit in its own regard (and will help bring a more focused audience to the paper).

4) What are the mock injections performed as controls? Cas9 will introduce DSB-related stress, so adding a control such as mutating an unrelated (i.e. pigment) gene might be a better control for the experiments than just mock injections (with buffer?).

5) The authors gauge an AV-block by visual inspection (line 204). This is a highly complex electrophysiological phenotype and issue that requires calcium measurements and imaging/measurement of the cardiac syncytium's conductivity behavior. While reminiscent of an AV-block, the authors are encouraged to tone down their call of this particular phenotype.

* Minor Points:

1) Introduction: "markers" should possibly rephrased to "causative genes" or the likes.

2) The statement that genes without pre-existing evidence are not being characterized seems once more superficial and not what prior publications show. Suggest rephrasing.

3) That the heart rate of fishes is more in line with the human heart rate (while the entire structure of the heart with two vs four chambers is entirely different and the pacemaker system is only rudimentary similar) seems like a weak argument to promote fishes (in this case particularly medaka) as model compared to mice. Again, the authors don't need to bolster their model - it is valid and interesting as it is.

4) cmlc2 should be renamed to the contemporary nomenclature, myl7.

5) Was the targeting of the GFP transgene (line 82) performed on heterozygous, i.e. single-copy, embryos? That should be stated or clarified in relation to the need of biallelic mutagenesis of endogenous loci.

6) Throughout the text, the authors are encouraged to add n=x data already into the text to solidify their claims and statements.

7) Line 225 and other instances: cardiovascular diseases can be prevented if diagnosed early, but this mostly concerns acquired issues (blood pressure, clotting issues, etc.). The aeuthors' assay is more suited to uncover congenital issues affecting structure and overall function of the heart. Revisiting these statements throughout is recommended.

8) Line 244 ff is a prime example for how the authors omit prior work (unnecessarily so) by discussing prior zebrafish work without providing a single reference. Similarly line 254, while possibly true for Medaka, lots of prior refinements in using RNPs, salt-based solubilization, etc. has been done in other models.

9) The authors argue that AI is required for deeper phenotyping of heart issues. The reviewer would argue that decades imaging and developmental biology work has shown pretty well that grad students, postdocs, and other lab personnel with sufficient training and expertise can manage to decode cardiac phenotypes. This should be rephrased or further explained, i.e. how AI would actually help (i.e. in combo with high-throughput imaging, etc.).

6. PLOS authors have the option to publish the peer review history of their article (what does this mean?). If published, this will include your full peer review and any attached files.

Reviewer #1: No

Reviewer #2: No

---

## [Author Response · Author response to Decision Letter 0]

28 Oct 2021

Dear Dr Varshney,

we have been very pleased by the constructive comments and advice provided by the referees and have revised our manuscript accordingly. In particular we have introduced additional data underlining the efficacy of the validation pipeline. We are confident that in our rebuttal letter and the associated changes we have been addressing all of the points raised by the referees and are looking forward to your and their response.

sincerely

Jochen Wittbrodt

---

## [Decision Letter · Decision Letter 1]

6 Dec 2021

*In vivo* identification and validation of novel potential predictors for human cardiovascular diseases

PONE-D-21-24196R1

Dear Dr. Wittbrodt,

We’re pleased to inform you that your manuscript has been judged scientifically suitable for publication and will be formally accepted for publication once it meets all outstanding technical requirements. 

I agree with the reviewer the authors have done a terrific job in revising this manuscript. Reviewer 1 has a minor point regarding the orientation of images in Figure 1 and others, I will appreciate if the authors could consider to revise.

Kind regards,

Gaurav Varshney, Ph.D.

Academic Editor

PLOS ONE

Reviewers' comments:

Reviewer's Responses to Questions

**Comments to the Author**

1. If the authors have adequately addressed your comments raised in a previous round of review and you feel that this manuscript is now acceptable for publication, you may indicate that here to bypass the “Comments to the Author” section, enter your conflict of interest statement in the “Confidential to Editor” section, and submit your "Accept" recommendation.

Reviewer #2: All comments have been addressed

2. Is the manuscript technically sound, and do the data support the conclusions?

Reviewer #2: Yes

3. Has the statistical analysis been performed appropriately and rigorously? 

Reviewer #2: I Don't Know

4. Have the authors made all data underlying the findings in their manuscript fully available?

Reviewer #2: Yes

5. Is the manuscript presented in an intelligible fashion and written in standard English?

Reviewer #2: Yes

6. Review Comments to the Author

Reviewer #2: The authors have done a laudable job in addressing the raised points. The reviewer is looking forward to seeing the manuscript in print/online.

One possibly rather minor point: the heart imaging shows the ventricle on the right in some images, and on the left in others (i.e. Fig 1 vs subsequent figures). The authors should make sure to point out which images are ventral vs dorsal views (and ideally also not mirrored images through the confocal setup) to also support readers more used to the stereotypic zebrafish heart's anatomy (which the reviewer assumes will be the majority of the readers...).

7. PLOS authors have the option to publish the peer review history of their article (what does this mean?). If published, this will include your full peer review and any attached files.

Reviewer #2: No

---

## [Editor Report · Acceptance letter]

9 Dec 2021

PONE-D-21-24196R1 

*In vivo* identification and validation of novel potential predictors for human cardiovascular diseases 

Dear Dr. Wittbrodt:

I'm pleased to inform you that your manuscript has been deemed suitable for publication in PLOS ONE. Congratulations! Your manuscript is now with our production department. 

Kind regards, 

on behalf of

Dr. Gaurav Varshney 

Academic Editor

PLOS ONE